# The Superiority of Fibroblast Activation Protein Inhibitor (FAPI) PET/CT Versus FDG PET/CT in the Diagnosis of Various Malignancies

**DOI:** 10.3390/cancers15041193

**Published:** 2023-02-13

**Authors:** Yanzhao Dong, Haiying Zhou, Ahmad Alhaskawi, Zewei Wang, Jingtian Lai, Chengjun Yao, Zhenfeng Liu, Sohaib Hasan Abdullah Ezzi, Vishnu Goutham Kota, Mohamed Hasan Abdulla Hasan Abdulla, Hui Lu

**Affiliations:** 1Department of Orthopedics, The First Affiliated Hospital of Zhejiang University, #79 Qingchun Road, Hangzhou 310003, China; 2School of Medicine, Zhejiang University, #866 Yuhangtang Road, Hangzhou 310058, China; 3Department of Nuclear Medicine, The First Affiliated Hospital of Zhejiang University, #79 Qingchun Road, Hangzhou 310003, China; 4Department of Orthopaedics, Third Xiangya Hospital of Central South University, #138 Tongzipo Road, Changsha 410013, China; 5Alibaba-Zhejiang University Joint Research Center of Future Digital Healthcare, Zhejiang University, #866 Yuhangtang Road, Hangzhou 310058, China

**Keywords:** FAPI, PET/CT, imaging, cancers, diagnosis

## Abstract

**Simple Summary:**

Fibroblast activation protein is a newly discovered biomarker in various tumor types. Using inhibitors ligated with radiopharmaceuticals, nuclear medicine physicians can identify primary tumors, lymphatic spread, and distant metastases. This review mainly summarizes the current application of the mentioned imaging method in many tumor types, with a comparison to the existing radiological diagnostic tools.

**Abstract:**

Cancer represents a major cause of death worldwide and is characterized by the uncontrolled proliferation of abnormal cells that escape immune regulation. It is now understood that cancer-associated fibroblasts (CAFs), which express specific fibroblast activation protein (FAP), are critical participants in tumor development and metastasis. Researchers have developed various FAP-targeted probes for imaging of different tumors from antibodies to boronic acid-based inhibitor molecules and determined that quinoline-based FAP inhibitors (FAPIs) are the most appropriate candidate as the radiopharmaceutical for FAPI PET/CT imaging. When applied clinically, FAPI PET/CT yielded satisfactory results. Over the past few years, the utility and effectiveness of tumor detection and staging of FAPI PET/CT have been compared with FDG PET/CT in various aspects, including standardized uptake values (SUVs), rate of absorbance and clearance. This review summarizes the development and clinical application of FAPI PET/CT, emphasizing the diagnosis and management of various tumor types and the future prospects of FAPI imaging.

## 1. Introduction

Cancer has plagued humanity for thousands of years and has become the second leading cause of death worldwide, behind cardiovascular diseases [1]. Cancer can occur in various organs and is characterized by the uncontrolled proliferation of abnormal cells, which then escape immune regulation and eventually metastasize throughout the body, causing loss of function in the affected organs, cancer pain, systematic failure, and death. Additionally, cancers can also cause impingement and compression on the adjacent nerves, resulting in pain and loss of motor and sensory functions in the innervated area.

The etiology of cancer development is complex and involves various microenvironment events. In 1889, Stephen Paget proposed the “seed and soil” theory, describing the cancerous cells as seeds and the adjacent tissue/organ as soil [2]. It is now understood that the tumor microenvironment (TME) is highly related to the origin and development of cancer. The TME is an ecosystem of non-oncogenic components within and adjacent to the tumor tissue, including cancer-associated fibroblasts (CAFs), the extracellular matrix (ECM), and various immune cells. Typically, TME can generate islands of immunosuppression in which tumor cells are shielded from immunological surveillance and circulating medication, allowing tumors to resist treatment and proliferate uncontrollably, deterring conventional anti-tumor treatment [3].

### 1.1. Cancer-Associated Fibroblasts (CAFs)

CAFs represent some of the most common cells in TME and are a key component in the TME, exerting a wide range of physiological and pathological functions in most solid tumors, including the deposition and remodeling of ECM [4]. CAFs also play important roles in the reciprocal cell-to-cell interactions among normal and cancerous cells, as well as the interaction and crosstalk among the infiltrating leukocytes and other components responsible for the surveillance and activity of the immune system. On the other hand, CAFs are capable of directly or indirectly regulating several tumoral pathological phenomena, such as tumor initiation, neovascularization, and metastasis [5,6]. Moreover, CAFs have the potential to participate in cell communication under pathological and physiological conditions by secreting multiple chemokines and cytokines including transforming growth factor-β, CC-chemokine ligand 2, and interleukin-6, which could recruit immunocytes in the tumor stroma while forming a stromal niche and exert inhibitory effects, hence facilitating immune evasion [7]. CAFs are well established to exhibit both tumor-promoting and tumor-suppressive effects under different circumstances [8]. Current evidence suggests that this heterogeneous population of cells expresses several biomarkers, including α-smooth muscle actin, fibroblast activation protein (FAP), and platelet-derived growth factor receptor-β [9]. At present, our understanding of the origin of CAFs and the heterogeneity in CAF function remains limited. Given the anti-tumorigenic functions of CAFs, further research is required for their future application in the clinical diagnosis and management of solid tumors.

### 1.2. Fibroblast Activation Protein and Fibroblast Activation Protein Inhibitor (FAPI)

FAP, a type II integral membrane glycoprotein, has been identified as an important biomarker in CAFs. As a member of the serine protease kinase family, FAP is involved in ECM remodeling and fibrogenesis. While the expression level of FAP is low to non-detectable in normal organs, it is greatly upregulated at sites of tissue remodeling and tumor stroma [10]. Importantly, FAP is involved in various pathological processes and can be detected in more than 90% of malignant epithelial tumors [11,12]. One of the most promising prospects associated with targeting FAP is the application of molecular imaging technologies, including positron emission tomography (PET) and single-photon emission computed tomography (SPECT), which is based on the development of FAP inhibitors exhibiting rapid and almost complete internalization upon administration and rapid clearance from the circulation [13,14,15]. It is widely believed that FAP expression is associated with worse prognosis in colorectal, pancreatic, ovarian, and hepatocellular cancer [11].

The first clinical application of FAP-based imaging involving the use of antibodies and boronic acid-based inhibitor molecules demonstrated limited clinical value given the low blood clearance rate and limitations associated with gamma imaging (Figure 1) [16,17,18]. More recently, quinoline-based FAPIs have been developed as radiopharmaceuticals for PET/CT imaging [19,20]. As a novel PET/CT tracer, FAPIs were compared with 18F-fluorodeoxyglucose (18F-FDG) in tumor detection rate and clinical efficiency, showing a higher sensitivity in identifying primary tumors and metastases involving lymph node, bone, and viscera [21,22,23,24,25,26].

## 2. Application of FAPI PET/CT in the Diagnosis and Treatment of Cancers

The novel imaging agent FAPI is currently applied during clinical practice to diagnose and manage several malignant tumors and their associated metastases. In this respect, Jiang et al. and Kuten et al. have observed a higher tumor-to-background ratio (TBR) in FAPI than in FDG, with little variations and fluctuations in the SUVmax in ovarian cancers and colorectal cancers [27,28]. In this review, the application of FAPI imaging in diagnosing and managing major types of tumors is discussed for future research and application of FAPI PET/CT (Figure 2).

### 2.1. Gastrointestinal Cancer

#### 2.1.1. General Background, Gastric Cancer, and Peritoneal Cancer

Gastric cancer in humans can be histologically categorized as diffuse and intestinal-type carcinoma according to the Lauren classification [29]. Although both types express high levels of FAP, higher levels have been documented in the intestinal type. In gastric carcinoma, FAP is majorly confined within carcinoma cells, which is different from many other tumor types [30,31]. Studies have shown that elevated FAP expression is associated with higher tumor grades, increased incidence of lymph node and lymphatic spread, peritoneal invasion, poor prognosis, and worse overall survival [32,33]. Gastric cancer models and related research have observed increased levels of ex vivo proliferation and migration of gastric carcinoma cells and elevated in vivo resistance against anti-PD1 therapy in co-cultures with FAP-expressing fibroblasts, and increased tumor size in vivo [33,34]. Compared with the traditional 18F-FDG PET/CT, the sensitivity of 68Ga-FAPI-04 PET/CT is significantly higher in detecting primary gastric tumors, positive lymph node and lymphatic metastasis, and distant metastases, especially in signet-ring cell carcinomas and mucinous adenocarcinomas [35]. Studies have reported the superiority of FAPI over FDG as tracers for PET/CT by measuring the SUVmax during gastric cancer detection and diagnosis [36].

#### 2.1.2. Colorectal Cancer

While FAP is ubiquitously expressed at a low to non-detectable level in normal tissue, high levels of FAP have been detected in human colorectal cancer cells and adjacent stroma, even at early stages [37,38]. Researchers have recently proposed that FAP plays a potential role in colorectal tumor migration or invasion since elevated levels of FAP have been detected in the tumor margin compared with the tumor center, and the pattern is reversed post-radiotherapy, indicating a role of FAP in tumor remodeling following radiological damage [39]. It has been established that an elevated level of FAP is associated with more advanced invasion depth, lymph node metastases, stage of the tumor, and, consequently, worse prognosis and overall survival [11,39]. On the other hand, FAP expression in tumors is associated with altered immunological traits, especially in the immune cell populations. For instance, elevated FAP is correlated with increased CD11b+ cell expression and decreased CD3+ cell expression [5]. Importantly, 68Ga-DOTA-FAPI-04 PET/CT showed similar sensitivities during the detection of primary colorectal tumors compared to FDG imaging while displaying higher SUVmax in the detection of lymph node metastases [40]. 

#### 2.1.3. Hepatobiliary and Pancreatic Carcinoma

As one of the common clinical malignancies, hepatobiliary carcinomas affect over 900,000 people globally, posing a serious challenge for families and healthcare professionals [41]. A recent study by Zou et al. revealed that the expression of FAP was induced by hypoxia and correlated with poor clinical outcomes. Furthermore, it was determined that the expression levels of FAP and HIF-1α were closely correlated via immunohistochemistry analysis, Western blot and qRT-PCR analysis conducted on a 138-patient cohort [42]. Regarding hepatobiliary malignancies, FAPI-based imaging methods, especially 68Ga-FAPI-04 PET/CT, demonstrated better sensitivity and accuracy in detecting the primary tumor, lymph node metastases, and distant metastases in comparison to FDG-based imaging systems (Table 1.) [43]. In the case of hepatocellular carcinomas and intrahepatic cholangiocarcinoma, FAPI-04 has demonstrated better performance in identifying primary liver tumors and extrahepatic metastases with higher tumor uptake and lower hepatic background uptake compared with 18F-FDG [14,44,45,46].

Positive FAP staining is observed in 90% of human pancreatic ductal adenocarcinomas (PDAC), while PDAC tumor cells and stromal cells were positively stained in cultured cell lines, with the stroma adjacent to the tumor tissue showing the highest levels of FAP [6]. Current evidence suggests that low FAP expression correlates with increased pancreatic fibrosis, and high FAP expression is generally associated with adverse events, including lymphatic metastases, tumor recurrence, and mortality [47,48]. Interestingly, it has been documented that FAP-positive pancreatic stellate cells underwent CXCL1-mediated Akt phosphorylation, subsequently enhancing tumor migration and invasion [34]. A study showed that in vivo FAP deficiency could delay tumor onset and prolong the lifespan of mice, and FAP depletion or deficiency increased tumor necrosis and impeded metastasis, especially at distant sites [48]. Other murine experiments using the KPC mice model of human PDAC have also observed increased FAP expression in the premalignant and malignant lesions [49]. Overall, these studies have demonstrated a positive correlation between elevated FAP levels and increased risk of poor prognosis [6,48]. While many believe that FAP is correlated with worsened clinical outcomes, clinicians are endeavoring to leverage these characteristics of FAP in anti-tumor therapeutic regimens. For instance, FAP blockade in combination with radiotherapy against pancreatic adenocarcinoma in a murine model has shown satisfactory outcomes, which led to the conclusion that anti-tumor T cell infiltration and function in pancreatic adenocarcinoma is enhanced via this regimen [50]. However, the tumor clearance rate by the mentioned method was insufficient to improve prognosis or extend survival. These results call for more research into FAP/FAPI and its role in PDAC, and pancreatic cancer in general [47].

**Table 1 cancers-15-01193-t001:** Comparison of FAPI PET/CT and FDG PET/CT in the imaging of hepatobiliary and gastrointestinal carcinomas. HCC: hepatocellular carcinoma; ICC: Intrahepatic cholangiocarcinoma; PC: pancreatic carcinoma; GC: gastric carcinoma; DC: duodenal carcinoma; CC: Colorectal carcinoma.

**Tumor Type**	**No. of Patients**	**Clinical Setting**			**FAPI Patient Analysis**				**FAPI Lesion Analysis**				**FDG Patient Analysis**				**FDG Lesion Analysis**				**Ref.**
		**Staging**	**Restaging**	**Treatment Status**	**TPR**	**FPR**	**TNR**	**FNR**	**TPR**	**FPR**	**TNR**	**FNR**	**TPR**	**FPR**	**TNR**	**FNR**	**TPR**	**FPR**	**TNR**	**FNR**	
HCC and ICC	34	50%	26%	23 treatment-naïve	96%	0	100%	4%	87%	0	100%	13%	65%	0%	100%	35%	65%	0%	100%	35%	Guo, W. et al. [44]
HCC and ICC	27	22%	67%	6 treatment-naïve	100%	7%	93%	0	96%	0	100%	4%	58%	0	100%	42%	89%	0	100%	11%	Siripongsatian, D. et al. [51]
PC	36	53%	26%	23 treatment-naïve	100%			0	82%	14%	86%	18%	52%			48%	59%	19%	81%	41%	Pang, Y. et al. [52]
GC, CC and DC	19	54%	46%	Treatment-naïve	100%			0	89%			11%	71%			39%	57%			43%	Pang, Y. et al. [53]
GC	56	80%	20%	45 treatment-naïve	100%			0	98%			2%	98%			2%	96%			4%	Lin, R. et al. [54]

### 2.2. Brain Cancer 

Although FAPI PET/CT has been extensively implemented in brain cancer research, the staging and subsequent treatment of brain cancers remain a significant conundrum, especially the determination of the tumor margins and invasion status. Glioma is the most common primary intracranial tumor with many subtypes. According to the World Health Organization, glioma can be categorized into isocitrate dehydrogenase (IDH) wild type and IDH mutant glioma types. The former type has been used to investigate the target affinity and specificity of FAPI-02 and FAPI-04 in animal studies, where 68GaFAPI-02/04 was used based on PET images. FAPI PET imaging showed a decreased tracer uptake in diffuse astrocytomas. On the other hand, increased tracer uptake by IDH-wildtype glioblastomas and high-grade IDH mutant astrocytomas was observed. PET images showed that FAPI-02 exhibits lower tumor accumulation and earlier elimination than FAPI-04 [55].

Lumbosacral myxopapillary ependymoma is a rare type of tumor that predominantly involves the conus medullaris, filum terminale, or cauda equina in the central nervous system. A recent study by Chongjia Li et al. revealed the diagnostic value of 68Ga-FAPI-04 PET/CT imaging in the current clinical settings [56]. 18F-FDG and 11C-MET showed a characteristic accumulation in high-grade ependymoma but brought little benefit in differentiating low-grade ependymoma from the background or non-neoplastic lesions. On the other hand, using 68Ga-FAPI-04 PET/CT imaging, low background uptake in FAPI was observed in the brain and spinal cord, which is suitable for the imaging of low-grade ependymoma. Other studies substantiated the diagnostic and management value of FAPI PET/CT imaging in primary benign intraosseous meningioma and schwannomas [57,58].

### 2.3. Neck and Ear, Nose, and Throat (ENT) Cancer

Head and neck cancers (HNCs) are among the most common malignancies worldwide, with more than half a million cases and a quarter million deaths reported annually. As an important part of traditional HNCs management, radiation therapy (RT) can significantly improve the overall prognosis and survival of patients. Despite its advantages, the shortcomings of RT include tumor recurrence, especially at the tumor margins, and toxicity to normal surrounding tissues [59,60]. The invasive and diffuse nature of HNCs and their tendency to infiltrate the surrounding tissue make diagnosis and treatment extremely challenging since conventional CT and MRI imaging exhibit poor ability to provide high-resolution images to contrast the tumor and surrounding tissues [60,61,62,63]. FDG PET/CT has been extensively applied for HNCs staging and treatment since its early application in 2015 by Schwartz D. et al. [64]. However, the application of FDG PET/CT is limited because of the high uptake rate of FDG by normal surrounding tissues, including muscles, the brain, salivary glands, and lymph nodes, which could result in a blurry background and the inability to differentiate tumors from inflamed or post-operative tissue [60]. Importantly, FAPI expression in HNCs is rarely influenced by the metabolic activity of the surrounding tissues, overcoming the limitations of FDG imaging. 

The application of FAPI PET/CT in detecting HNCs has become a research hotspot in recent years. It has been shown that FAPI PET/CT could facilitate precision radiotherapy [22,65,66,67]. Indeed, it is widely acknowledged that FAPI PET/CT targets the HNCs more accurately and sensitizes the whole tumor to RT and chemoimmunotherapy via targeting CAF-specific cell surface markers, CAF-specific paracrine signaling pathways, and CAF-derived ECM. Given that FAP expression is exclusively high in primary lesions and metastases, harm and toxicity to the surrounding tissues are limited [68,69]. Manuel et al. measured the 68Ga-FAPI PET/CT uptake by adenoid cystic carcinoma (ACC) and determined that 68Ga-FAPI PET/CT identifies more metastatic lesions compared with CT or MRI, which helps in upstaging the ACC and improving the RT by increasing the accuracy of the target gross tumor volume (GTV) delineation confirmed by SUVmax and TBR [70]. In a pilot study on the delineation of HNCs for RT planning, M. Syed et al. investigated an existing database of 14 HNC patients using both CE-CT for conventional GTVs and 68Ga FAPI PET/CT for FAPI-GTVs, which was later analyzed with syngo. via software. An improved quality of imaging was obtained, and it was concluded that a greater initial dosage was required for RT planning due to the enhanced detection ability of lymph node metastases and distant metastases [71]. Chunxia et al. compared the value of 68Ga-FAPI PET/MR and 18F-FDG PET/MR in diagnosing and staging of nasopharyngeal carcinoma (NPC) patients and observed that 68Ga-FAPI PET/MR was superior in evaluating intracranial invasion, skull base, and the small metastases [72]. However, FAPI PET/CT still warrants further improvement. In this respect, a recent case report revealed that FAPI PET/CT yielded repeated false positive results in a patient suffering from esophagitis, possibly due to its inability to differentiate between cancerous tissue and inflammation [73].

On the other hand, in 2022, it was reported that FAPI-guided targeted therapy yielded a substantial effect in a patient suffering from aggressive medullary thyroid carcinoma. Importantly, this was the first-in-human attempt of 68Ga-DOTA.SA.FAPI–guided 177Lu-DOTAGA, a novel combination of radiopharmaceuticals with FAPI imaging modalities [74]. This success corroborates that FAPI imaging can be harnessed to guide the treatment of aggressive medullary thyroid cancer and other HNCs. 

### 2.4. Lung Cancer

18F-FDG PET/CT is now recommended by the National Comprehensive Cancer Network Guidelines, along with contrast-enhanced brain MRI, as the standard evaluation method for clinically staging lung cancer pre- and post-treatment [75,76]. However, concerns have been raised regarding its compromised ability to detect bone and pleural metastases of lung cancer [77,78]. To resolve this issue, Kratochwil C. et al. and Loktev A. et al. subsequently introduced 68Ga-FAPI PET/CT to compensate for these defects caused by the imaging principle of FDG, which is metabolic imaging [13,23]. Current evidence suggests that 68Ga-FAPI PET/CT is superior to 18F-FDG PET/CT in diagnosing and staging lung cancer, and it has already been applied clinically for detecting early-stage lung cancer. However, it should be noted that 68Ga-FAPI PET/CT is not a routine examination for diagnosing the primary lesions or metastases of patients with lung cancers [79]. Furthermore, some studies compared the two imaging modalities regarding primary lung cancers and suspected liver or adrenal gland metastases from lung cancer. It was determined that 68Ga-FAPI PET/CT detects more metastatic lesions in the liver, lungs, pleura, bone, and brain, and the SUVmax and TBR values were higher in bone and hepatic lesions with 68Ga-FAPI PET/CT [80]. Furthermore, Giesel L. et al. and Kratochwil C. et al. demonstrated that 68Ga-FAPI PET/CT could more easily detect mediastinal lymph node metastasis with higher uptake and TBR [22,23].

In contrast with FDG PET/CT, 68Ga-FAPI PET/CT has low background radioactivity in the brain given that normal brain tissue only expresses low to non-detectable FAP levels while the metabolic rate is high, resulting in high background 18F-FDG uptake and adding to the difficulty in differentiating tumor from normal tissue. Therefore, FAPI imaging is more sensitive in detecting brain and leptomeningeal metastases from lung adenocarcinoma (ADC) [81]. 

It has been established that FAPI PET/CT could differentiate between various types of lung cancer (squamous cell carcinoma SCC, ADC, non-small cell lung cancer NSCLC, and small cell lung cancer) as primary tumors or metastatic lesions in different sites [82]. A retrospective study was conducted to estimate and compare the diagnostic accuracy between 18F-FDG and 68Ga-FAPI-04 PET/CT in NSCLC patients, which showed a better diagnostic accuracy of 68Ga-FAPI-04 PET/CT based on TNM staging, while there is no significant difference in SUV max, TBR values for the primary tumor [80,82,83,84]. Contrasting 68Ga-FAPI PET/CT studies have reported higher primary tumor SUV max than 18F-FDG, which may be attributed to the heterogeneity in clinical settings and more advanced tumor staging [85]. 

Another advantage of FAPI imaging is that 12 min post-injection of 68Ga-FAPI could achieve a higher SUV, providing a better-quality evaluation image of lung cancer at an earlier time, which limits the risk of deleterious side effects induced by prolonged circulating radiopharmaceuticals often observed with 18F-FDG [86]. In addition, the probability of false-negative results by 18F-FDG has been reported to be higher when no 18F-FDG uptake was detected in the lymph node of less than 10 mm in size or of small volume and low metabolic activity tumors, while 68Ga-FAPI PET/CT yielded correct diagnoses under such conditions [87,88,89]. 

However, benign pulmonary diseases may produce false-positive results during 18F-FDG PET/CT, such as tuberculosis, silicosis, and sarcoidosis [87,90,91]. For instance, Canan C. et al. reported false-positive results by 18F-FDG due to anthracosis in a 29-patient cohort of patients suffering from NSCLC, ADC, and SCC, where 18F-FDG and 68Ga-FAPI yielded 89.6% and 100% diagnostic accuracy, respectively [80]. While FAPI imaging is superior to FDG in diagnostic accuracy, it warrants further improvement, especially for differentiating between cancer and other pathological conditions including infection, inflammation, and fibrosis [92].

### 2.5. Breast Cancer 

Targeting FAP has become a promising breast cancer imaging and treatment approach in recent years. In a cohort of 48 breast cancer patients, 68Ga-FAPI-04 PET/CT displayed superior diagnostic value in the number of detected lesions of primary tumors with higher SUVmax, lymphatic metastases, and distant metastases than 18F-FDG PET/CT [93]. Concerning the management of breast cancer, one of the earliest studies on FAP expression and relevance in breast cancer by Thomas Kelly et al. showed that FAP-α promotes tumor growth and invasion of breast cancer cells via non-enzymatic functions mediated by MMP-9 in a mouse model of human breast cancer [94].

Another early publication in this domain compared breast epithelial tumors with normal breast tissue and observed FAP overexpression in tumor stroma [95]. In 2011, Hua X. et al. proposed that the application of FAP staining resulted in the up-staging of ductal carcinoma in situ (DCIS) to DCIS with microenvironment invasion (DCIS-MI), which could facilitate fine-tuning of therapeutic management associated with DCIS and DCIS-MI [96].

Current evidence suggests that FAP expression in the stroma is higher in cases of invasive lobular carcinoma than in invasive carcinoma with no special characteristics or type [97]. In invasive ductal carcinoma, high-grade tumors and stroma with inflammation and adipocytes had higher FAP levels than those with desmoplastic, sclerotic, or normal-like stroma [98]. In benign breast tumors such as phyllodes tumors, higher levels of FAP mRNA correlated with malignant transformation, suggesting that, in addition to its prognostic value in DCIS, FAP represents a potential biomarker signaling malignancy [99].

### 2.6. Bone Tumors 

Regarding potential applications in the musculoskeletal system, 68Ga-FAPI-04 PET/CT is mainly used to diagnose bone metastases of malignant tumors originating elsewhere rather than primary bone malignant tumors. Bone metastasis is a common complication in patients suffering from advanced malignant tumors involving the breast, prostate, lung, thyroid, kidney, etc. [100,101]. Indeed, early detection and timely treatment of bone metastases can significantly improve the quality of life and overall prognosis by avoiding complications ranging from cancer-associated osteoporosis to pathological fracture [102]. There is an increasing consensus that compared with 18F-FDG PET/CT, FAPI PET/CT yields a higher detection rate of bone metastases with high sensitivity, high uptake values, and high TBR [53,92,103]. Chen et al. showed that 68Ga-FAPI-04 PET/CT is superior to 18F-FDG PET/CT in detecting bone metastasis with significantly increased uptake by metastatic lesions (*n* = 9, median SUV 9.3 vs. 2.6, *p* = 0.008) [104]. These conclusions were corroborated by a retrospective analysis conducted by Wu et al., where a cohort of 75 patients underwent 68Ga-DOTA-FAPI-04 PET/CT and 18F-FDG PET/CT, and the sensitivity for bone and visceral metastases was 83.8% vs. 59.5% (*p* = 0.004) [92]. 

However, false-positive results in benign bone lesions 68Ga-FAPI PET/CT have been reported, as observed in a case of a 72-year-old patient suffering from myositis ossificans, a benign orthopedic disorder [105]. Zheng et al. reported that both 18F-FDG and 68Ga-FAPI imaging led to a false-negative diagnosis of bone metastasis in a breast cancer patient, which is also the first case of a false-negative result of bone metastasis by FAPI imaging [106]. Accordingly, it is premature to conclude that 68Ga-FAPI is superior to 18F-FDG in detecting bone metastases or lesions, emphasizing the importance of further research.

### 2.7. Soft Tissue Tumors

Current research on the applications of FAPI PET/CT in cancer confirmed that FAPI led to high SUV values in sarcomas and distinct contrast to the background due to low background uptake [15,23,107,108]. Regarding the evaluation of diagnostic accuracy, 68Ga-FAPI PET/CT yielded a high positive predictive value in patient-based analysis (*n* = 27, 100%) and lesion-based analysis (*n* = 33, 97%), as well as high sensitivity (*n* = 28, 96%; *n* = 34, 94%) [109]. In addition, FAPI imaging has huge prospects to guide therapy against sarcomas given that various FAPI uptake rates have been observed at different stages of sarcomas [110,111]. In this respect, a recent study by Koerber et al. on seven types of sarcomas in 15 patients showed high uptake in FAPI in high-grade sarcomas, including undifferentiated pleomorphic sarcoma (UPS) and low uptake in low-grade sarcomas, which could be applied in RT [108]. Although 18F-FDG PET/CT is highly effective in diagnosing primary lesions, especially in high-grade sarcomas, FAPI is considered to possess superior diagnostic value, as shown in studies conducted by Macpherson et al. (*n* = 493), Charest et al. (*n* = 212) [112,113,114]. Studies by Bingxin Gu et al. have shown that FAPI is superior to 18F-FDG PET/CT (*n* = 45) in identifying recurrent lesions of liposarcoma, malignant solitary fibrous tumors, and invasive ductal carcinomas in sensitivity (97.52% vs. 65.96%), specificity (60.71% vs. 21.43%), positive predictive value (96.15% vs. 89.42%), negative predictive value (70.88% vs. 5.88%), and accuracy (94.19% vs. 61.94%), while on the other hand, 18F-FDG showed better diagnostic efficacy than FAPI PET/CT in identifying UPS and rhabdomyosarcoma [115]. Both imaging modalities yielded similar results in monitoring the recurrence of bone malignancies [115].

Apart from sarcomas, research on FAPI in detecting other soft tissue tumors, including malignant melanoma, remains relatively scarce [116]. Kgomotso Mokoala et al. presented a case of advanced cutaneous malignant melanoma with multiple organ metastasis, where FAPI yielded higher SUV values than FDG in the liver, lung, and left femoral head metastases but not for lung metastases [117]. However, in the case of a 65-year-old patient with metastatic uveal malignant melanoma, the SUV for FAPI was lower than FDG in the metastatic liver lesion while higher in both knee joints [118]. This discrepancy may result from the limited sample volume, highlighting the need for further investigation. 

### 2.8. Other Cancer

The significant heterogeneity of the cell/extracellular microenvironment is associated with the prognosis of patients with lymphoma [119]. Depending on different phenotypes, CAFs may exhibit pro-tumor or anti-tumor effects in the TME [120]. Hodgkin lymphoma (HL) is characterized by the proliferation of fibroblasts in the TME. Studies have confirmed that FAP immunostaining is more intense in neoplasms, and FAPI uptake is significantly increased in HL lesions. However, compared with HL, FAP immunostaining and FAPI uptake are weaker in NHL except for aggressive NHL, especially diffuse large B-cell lymphoma [121]. The high TBR in FAPI images resulted in high contrast ratios for lymphoma lesions, especially in the brain, liver, and oropharynx, which can be very helpful in diagnosing lymphoma in these organs [122,123].

## 3. Conclusions

FAPI and the related imaging techniques have greatly propelled the development of diagnosis and management schemes for malignant tumors involving various organs. Before the clinical application of FAPI imaging, the accurate and general imaging of malignant tumors and metastatic lesions were usually achieved by 18F-FDG imaging, which has been substantiated to be less effective than FAPI imaging in various clinical settings. This review provided a comprehensive overview of CAFs and TME, the basis for FAPI imaging, and several types of tumors where FAPI imaging has achieved great diagnostic and therapeutic potential. While limited by the number of patients included in many studies, it can be inferred that FAPI imaging could open new horizons in oncological diagnosis and management compared with conventional imaging and radiotherapy methods. 

However, FAPI imaging has been reported to produce false-positive results, possibly due to the inability to differentiate between cancerous and inflamed tissue. Indeed, more studies are warranted to validate the clinical advantages and disadvantages. In spite of the mentioned limitations, the clinical use of FAPI imaging and therapy is worthy of further research.

## Figures and Tables

**Figure 1 cancers-15-01193-f001:**
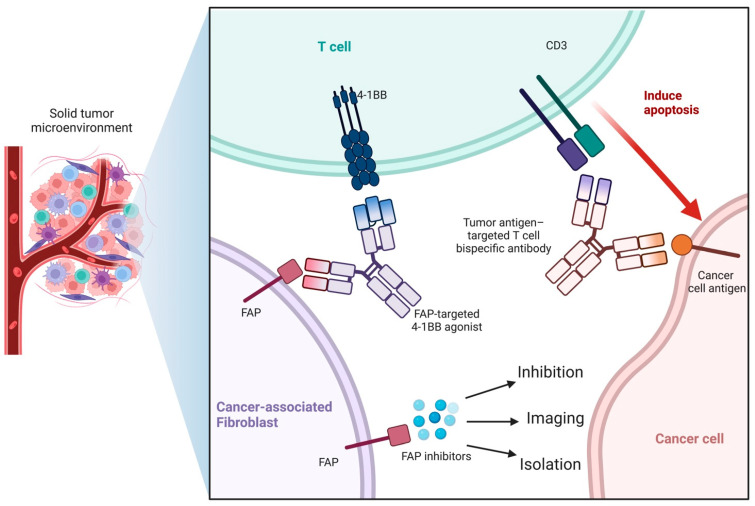
Interaction between FAP and FAPI in the solid tumor microenvironment. Tumor-targeted therapy could be achieved by FAP recognition and inhibition via 4-1 BB bispecific agonists, indicating the therapeutic value of FAPI.

**Figure 2 cancers-15-01193-f002:**
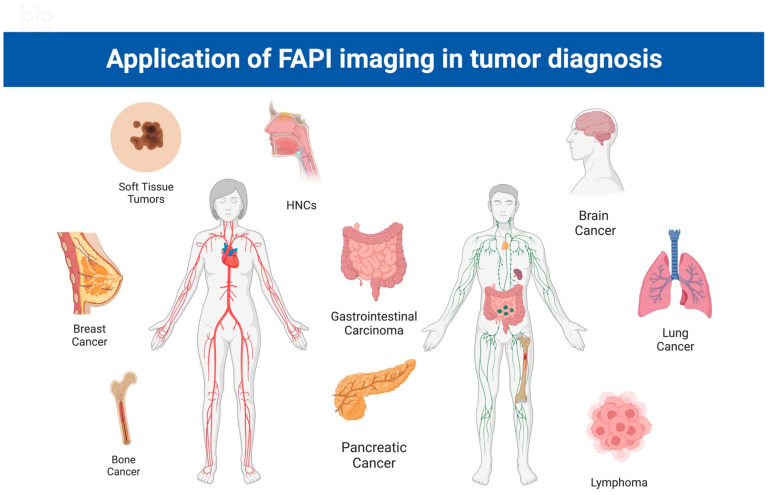
Application of FAPI imaging in various tumor diagnoses and management. HNC: head and neck carcinoma.

## Data Availability

The dataset supporting the conclusions of this article is included with the article.

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
