# Peer review of "The Superiority of Fibroblast Activation Protein Inhibitor (FAPI) PET/CT Versus FDG PET/CT in the Diagnosis of Various Malignancies"

_cancers, 2023, doi:10.3390/cancers15041193_

Round 1
Reviewer 1 Report
The aim of this review article is to summarize the clinical applications of FAPI PET/CT in cancer patients and compare the diagnostic performance of this tracer with that of 18F-FDG.
General comments
The manuscript provides a quite superficial view of FAPI PET/CT applications in various types of cancer without giving details on the sensitivity and specificity found in different studies and different cohorts of patients using this imaging modality. Therefore additional tables should be included containing for each study the number of patients, tumor type, clinical context (staging, restaging, RT planning), analysis per patient, analysis per lesion (true positive, true negative, false positive and false negative), eventually 18F-FDG PET/CT comparison.
English should be carefully revised since some sentences are unmeaningful. Repetitions should be avoided.
Specific comments
Introduction is very generic and elementary
Page 3 line 101. Please add all references on the initial use of 68Ga-FAPI such as Eur J Nucl Med Mol Imaging. 2019 Jul;46(8):1754-1755; J Nucl Med. 2019 Dec;60(12):1743-1749 and other papers published before the paper in Eur J Nucl Med Mol Imaging 2022, 49, 732-742.
Page 3 line 112. “ into such imaging tools” , what does this mean?
Figure 1. Legend should better explain the figure.
Page 4 line 119-120. This sentence is not clear
Please check the use of capitol letters in the headings of the different sections.
References 25-31 seem to be not appropriate for gastric cancer.
Page 4 lines 138-140. “An inverse-proportional relationship has been observed between the staining intensity of FAP and the tumor 139 stage, or the xenograft tumor size”. This sentence is incongruent with the content of the paragraph at page 4 lines 144-146.
Page line 150. It is not clear what the Heidelberg method is.
Reference 28 at page 5 line 152 is inappropriate.
Page 5 line 168. “One of Heidelberg method” is not clear what it is.
Page 5 line 174. “ex vivo cell lines” please change in “cultured cell lines”
Page 5 lines 192-194. This sentence should be reworded.
Table 1. Name and surname in the reference column are inverted. In this way authors in the table do not correspond to authors in the reference list.
Page 7 lines 244-245. “Several studies investigated the use of FAPI PET/CT in the detection and delineated that HNCs have accurate RT planning” This sentence should be reworded.
Page 7 lines 250-253. The statements in these two sentences are contradictory.
Page 8 line 290. “For a couple of years” should be perhaps “for a couple of decades”
Page 8 lines 316-320. This paragraph is a repetition.
Author Response
The manuscript provides a quite superficial view of FAPI PET/CT applications in various types of cancer without giving details on the sensitivity and specificity found in different studies and different cohorts of patients using this imaging modality. Therefore additional tables should be included containing for each study the number of patients, tumor type, clinical context (staging, restaging, RT planning), analysis per patient, analysis per lesion (true positive, true negative, false positive and false negative), eventually 18F-FDG PET/CT comparison.
Answer: Thank you for taking the time to review our manuscript. We have updated the table accordingly.
English should be carefully revised since some sentences are unmeaningful. Repetitions should be avoided.
Answer: we have thoroughly reviewed the manuscript and corrected grammatical and lexical errors.
Specific comments
Introduction is very generic and elementary
Answer: we have added to the content in the introduction and restructured it for clarity.
Page 3 line 101. Please add all references on the initial use of 68Ga-FAPI such as Eur J Nucl Med Mol Imaging. 2019 Jul;46(8):1754-1755; J Nucl Med. 2019 Dec;60(12):1743-1749 and other papers published before the paper in Eur J Nucl Med Mol Imaging 2022, 49, 732-742.
Answer: we have reviewed the mentioned literature and added to the reference.
Page 3 line 112. “ into such imaging tools” , what does this mean?
Answer: we have changed the wording for clarity.
Figure 1. Legend should better explain the figure.
Answer: we have added to the figure legend in figure 1.
Page 4 line 119-120. This sentence is not clear
Answer: we have restructured the sentence for clarity.
Please check the use of capitol letters in the headings of the different sections.
Answer: we have unified the format of headings of the different sections.
References 25-31 seem to be not appropriate for gastric cancer.
Answer: we have deleted several sentences to avoid repetition and aim the focus on gastric cancer.
Page 4 lines 138-140. “An inverse-proportional relationship has been observed between the staining intensity of FAP and the tumor 139 stage, or the xenograft tumor size”. This sentence is incongruent with the content of the paragraph at page 4 lines 144-146.
Answer: we have deleted lines 138-140 for clarity.
Page line 150. It is not clear what the Heidelberg method is.
Answer: we have specified the method.
Reference 28 at page 5 line 152 is inappropriate.
Answer: we have updated citation that is in accordance with our manuscript.
Page 5 line 168. “One of Heidelberg method” is not clear what it is.
Answer: we have deleted “one of Heidelberg method” for clarity.
Page 5 line 174. “ex vivo cell lines” please change in “cultured cell lines”
Answer: we have changed “ex vivo cell lines” to “cultured cell lines”.
Page 5 lines 192-194. This sentence should be reworded.
Answer: we have rewritten the sentence for clearance.
Table 1. Name and surname in the reference column are inverted. In this way authors in the table do not correspond to authors in the reference list.
Answer: we have altered the reference style in table 1 to improve matching between the table and the reference list.
Page 7 lines 244-245. “Several studies investigated the use of FAPI PET/CT in the detection and delineated that HNCs have accurate RT planning” This sentence should be reworded.
Answer: we have carefully restructured the sentence for clearance.
Page 7 lines 250-253. The statements in these two sentences are contradictory.
Answer: we have deleted the first sentence for clearance.
Page 8 line 290. “For a couple of years” should be perhaps “for a couple of decades”
Answer: we have changed the phrase to “for several decades”
Page 8 lines 316-320. This paragraph is a repetition.
Answer: we have deleted the repeated sentences for clearance.

Reviewer 2 Report
FAPI-PET is a very innovative method of molecular imaging. The topic is very "hot", because further development of this modern method, its application in scientific research and clinical management may revolutionize not only the current diagnostic procedures, but also, by learning the nature of many previously undescribed processes, the approach to treatment and management in many oncological diseases.
The authors of the publication prepared a review of the previous studies on FAPI-PET in cancer diagnosis. The review applies to all cancers. Certainly, the preparation of such a publication is not easy, because the available reports often cover small, heterogeneous groups or are only case reports. In manuscript, there is seen some kind of heterogeneity or inconsistency in the description of individual cancer types, which are sometimes described very briefly, or very practically and from various point of view: clinical use or as analysis of the basic processes occurring at the molecular level. It is difficult to say whether the publication is aimed more towards clinicians or preclinical scientists.
The manuscript shows numerous errors and shortcomings regarding the nomenclature, abbreviations used and the so-called "typos".
First of all, in bone tumors (2.6) and soft tissue tumors (2.7) we cannot write about "cancers", but about "lesions", "metastases" or "tumors".
Abbreviations are not used consistently, initially the authors use the abbreviation FAPI/PET/CT when writing about the method, then FAPI-PET/CT, FA-PI-PET/CT, FAPI PET/CT. This also applies to other abbreviations used. Some tumours names are capitalized, while others are lowercase. There are also abbreviations that are not explained. Conversely, some shortcuts once entered are not reapplied. The whole thing gives the impression that some care was lacking towards the end of the preparation of the manuscript.
Of course, the above errors do not affect the substantive quality of the work. However, there is a lack of information regarding the summarize of potential disadvantages of the described method and information on what they could result from.
Acceptance of the manuscript for publication should be considered after making corrections regarding the above-mentioned imperfections.
Author Response
The authors of the publication prepared a review of the previous studies on FAPI-PET in cancer diagnosis. The review applies to all cancers. Certainly, the preparation of such a publication is not easy, because the available reports often cover small, heterogeneous groups or are only case reports. In manuscript, there is seen some kind of heterogeneity or inconsistency in the description of individual cancer types, which are sometimes described very briefly, or very practically and from various point of view: clinical use or as analysis of the basic processes occurring at the molecular level. It is difficult to say whether the publication is aimed more towards clinicians or preclinical scientists.
Answer: Thank you for taking the time to review our manuscript. In our humble opinion, we hope that this review could be of help towards both clinicians and researchers in this field for future studies.
The manuscript shows numerous errors and shortcomings regarding the nomenclature, abbreviations used and the so-called "typos".
Answer: we have reviewed the manuscript and corrected several grammatical and lexical errors.
First of all, in bone tumors (2.6) and soft tissue tumors (2.7) we cannot write about "cancers", but about "lesions", "metastases" or "tumors".
Answer: we have corrected the mentioned errors.
Abbreviations are not used consistently, initially the authors use the abbreviation FAPI/PET/CT when writing about the method, then FAPI-PET/CT, FA-PI-PET/CT, FAPI PET/CT. This also applies to other abbreviations used. Some tumours names are capitalized, while others are lowercase. There are also abbreviations that are not explained. Conversely, some shortcuts once entered are not reapplied. The whole thing gives the impression that some care was lacking towards the end of the preparation of the manuscript.
Answer: we have carefully reviewed the manuscript and unified the style and abbreviation to improve the general quality.
Of course, the above errors do not affect the substantive quality of the work. However, there is a lack of information regarding the summarize of potential disadvantages of the described method and information on what they could result from.
Answer: we have included the potential disadvantage of FAPI and possible reasons.

Round 2
Reviewer 1 Report
General comments
The authors revised the manuscript without taking into account each point. Despite revision, some sentences remain unmeaningful and some passages appear obscure. The manuscript needs to be improved in the description of each study with an effort to report in the text the number of patients and main results in term of sensitivity and specificity. The manuscript needs also an extensive and careful english revision. References should be carefully checked.
Specific comments
Line 21 “radiologists” should be substituted with “nuclear medicine physicians”
Line 27 “is a central role”. This seems to be incorrect.
Line 29 “ac-id-based” should be “acid-based”
Line 114. The authors added the suggested references (Eur J Nucl Med Mol Imaging. 2019 Jul;46(8):1754-1755; J Nucl Med. 2019 Dec;60(12):1743-1749) in the wrong context. In fact these papers did not use antibodies and boronic acid-based inhibitor molecules but quinoline-based FAPI. In this respect, it could be helpful to remind that radiolabeled FAP inhibitors (FAPIs) for noninvasive imaging of FAP expression were developed and characterized by Haberkorn’s group including quinoline-based FAPI as reported in reference n. 15: Lindner T, Loktev A, Altmann A, Giesel F, Kratochwil C, Debus J, Jäger D, Mier W, Haberkorn U. Development of Quinoline-Based Theranostic Ligands for the Targeting of Fibroblast Activation Protein.J Nucl Med. 2018 Sep;59(9):1415-1422. doi: 10.2967/jnumed.118.210443
Furthermore, authors did not cite at line 119 all the other references related to papers published before the paper in Eur J Nucl Med Mol Imaging 2022, 49, 732-742 as for instance:
Giesel FL, Kratochwil C, Lindner T, et al. FAPI-PET/CT: biodistribution and preliminary dosimetry estimate of two DOTA-containing FAP-targeting agents in patients with various cancers. J Nucl Med. 2019;60:386–392;
Kratochwil C, Flechsig P, Lindner T, Abderrahim L, Altmann A, Mier W, Adeberg S, Rathke H, Röhrich M, Winter H, Plinkert PK, Marme F, Lang M, Kauczor HU, Jäger D, Debus J, Haberkorn U, Giesel FL. 68Ga-FAPI PET/CT: Tracer Uptake in 28 Different Kinds of Cancer. J Nucl Med. 2019 Jun;60(6):801-805. doi: 10.2967/jnumed.119.227967;
Giesel FL, Kratochwil C, Schlittenhardt J, Dendl K, Eiber M, Staudinger F, Kessler L, Fendler WP, Lindner T, Koerber SA, Cardinale J, Sennung D, Roehrich M, Debus J, Sathekge M, Haberkorn U, Calais J, Serfling S, Buck AL. Head-to-head intra-individual comparison of biodistribution and tumor uptake of 68Ga-FAPI and 18F-FDG PET/CT in cancer patients. Eur J Nucl Med Mol Imaging. 2021 Dec;48(13):4377-4385. doi: 10.1007/s00259-021-05307-1;
Koerber SA, Staudinger F, Kratochwil C, Adeberg S, Haefner MF, Ungerechts G, Rathke H, Winter E, Lindner T, Syed M, Bhatti IA, Herfarth K, Choyke PL, Jaeger D, Haberkorn U, Debus J, Giesel FL. The Role of 68Ga-FAPI PET/CT for Patients with Malignancies of the Lower Gastrointestinal Tract: First Clinical Experience. J Nucl Med. 2020 Sep;61(9):1331-1336. doi: 10.2967/jnumed.119.237016.
Dendl K, Koerber SA, Finck R, Mokoala KMG, Staudinger F, Schillings L, Heger U, Röhrich M, Kratochwil C, Sathekge M, Jäger D, Debus J, Haberkorn U, Giesel FL. 68Ga-FAPI-PET/CT in patients with various gynecological malignancies. Eur J Nucl Med Mol Imaging. 2021 Nov;48(12):4089-4100. doi: 10.1007/s00259-021-05378-0.
References 27-33 do not refer to gastric cancer. Please provide appropriate references on gastric cancer
Lines 194-197 Esophagus carcinoma is out of context here.
Reference numbers in table 1 are not correct.
Lines 275-277. This sentence is not clear.
Lines 290-291 “FDG PET/CT, as a newly developed imaging tool, is well known for HNCs staging and treatment”. Please reword this sentence.
Line 295 “HNCs under the CAFs category overexpress FAPI” What does under the CAF category mean?
Line 298-300 “HNCs has been extensively studied by several research groups” Please provide references.
Line 301-303 The study by Syed was incorrectly reported since this author studied 14 patients with head and neck cancer.
Lines 306-312. It is not clear how FAPI can sensitize whole tumor to RT. Also the sentence “FAPI PET/CT also improves surveillance of therapeutic management via dose painting, which is achieved based on the SUVs” is not clear.
Lines 321-324 Please reword this sentence.
Line 333-335. “68Ga-FAPI PET/CT surpasses 18F-FDG PET/CT in the diagnosis and staging of lung cancer and is applied in the detection of early-stage lung cancer”. Please specify that 68Ga-FAPI PET/CT is not currently included in the work-up of lung cancer patients.
Line 340-341 “SUVmax was similar but TBG was higher” Where?
In addition to the points raised, paragraphs 2.3, 2.4, 2.6 and 2.7 should be carefully revised to improve the content and the English. Also the concept of using PET images for delineation of gross tumor volume in radiotherapy planning should be better explained.
Author Response
General comments
The authors revised the manuscript without taking into account each point. Despite revision, some sentences remain unmeaningful and some passages appear obscure. The manuscript needs to be improved in the description of each study with an effort to report in the text the number of patients and main results in term of sensitivity and specificity. The manuscript needs also an extensive and careful english revision. References should be carefully checked.
Answer: Thank you for your time and effort to review this manuscript. We have had this manuscript reviewed thoroughly by native speakers and we are willing to provide proof along with the revised manuscript.
Specific comments
Line 21 “radiologists” should be substituted with “nuclear medicine physicians”
Line 27 “is a central role”. This seems to be incorrect.
Line 29 “ac-id-based” should be “acid-based”
Line 114. The authors added the suggested references (Eur J Nucl Med Mol Imaging. 2019 Jul;46(8):1754-1755; J Nucl Med. 2019 Dec;60(12):1743-1749) in the wrong context. In fact these papers did not use antibodies and boronic acid-based inhibitor molecules but quinoline-based FAPI. In this respect, it could be helpful to remind that radiolabeled FAP inhibitors (FAPIs) for noninvasive imaging of FAP expression were developed and characterized by Haberkorn’s group including quinoline-based FAPI as reported in reference n. 15: Lindner T, Loktev A, Altmann A, Giesel F, Kratochwil C, Debus J, Jäger D, Mier W, Haberkorn U. Development of Quinoline-Based Theranostic Ligands for the Targeting of Fibroblast Activation Protein.J Nucl Med. 2018 Sep;59(9):1415-1422. doi: 10.2967/jnumed.118.210443
Furthermore, authors did not cite at line 119 all the other references related to papers published before the paper in Eur J Nucl Med Mol Imaging 2022, 49, 732-742 as for instance:
Giesel FL, Kratochwil C, Lindner T, et al. FAPI-PET/CT: biodistribution and preliminary dosimetry estimate of two DOTA-containing FAP-targeting agents in patients with various cancers. J Nucl Med. 2019;60:386–392;
Kratochwil C, Flechsig P, Lindner T, Abderrahim L, Altmann A, Mier W, Adeberg S, Rathke H, Röhrich M, Winter H, Plinkert PK, Marme F, Lang M, Kauczor HU, Jäger D, Debus J, Haberkorn U, Giesel FL. 68Ga-FAPI PET/CT: Tracer Uptake in 28 Different Kinds of Cancer. J Nucl Med. 2019 Jun;60(6):801-805. doi: 10.2967/jnumed.119.227967;
Giesel FL, Kratochwil C, Schlittenhardt J, Dendl K, Eiber M, Staudinger F, Kessler L, Fendler WP, Lindner T, Koerber SA, Cardinale J, Sennung D, Roehrich M, Debus J, Sathekge M, Haberkorn U, Calais J, Serfling S, Buck AL. Head-to-head intra-individual comparison of biodistribution and tumor uptake of 68Ga-FAPI and 18F-FDG PET/CT in cancer patients. Eur J Nucl Med Mol Imaging. 2021 Dec;48(13):4377-4385. doi: 10.1007/s00259-021-05307-1;
Koerber SA, Staudinger F, Kratochwil C, Adeberg S, Haefner MF, Ungerechts G, Rathke H, Winter E, Lindner T, Syed M, Bhatti IA, Herfarth K, Choyke PL, Jaeger D, Haberkorn U, Debus J, Giesel FL. The Role of 68Ga-FAPI PET/CT for Patients with Malignancies of the Lower Gastrointestinal Tract: First Clinical Experience. J Nucl Med. 2020 Sep;61(9):1331-1336. doi: 10.2967/jnumed.119.237016.
Dendl K, Koerber SA, Finck R, Mokoala KMG, Staudinger F, Schillings L, Heger U, Röhrich M, Kratochwil C, Sathekge M, Jäger D, Debus J, Haberkorn U, Giesel FL. 68Ga-FAPI-PET/CT in patients with various gynecological malignancies. Eur J Nucl Med Mol Imaging. 2021 Nov;48(12):4089-4100. doi: 10.1007/s00259-021-05378-0.
Answer: we have corrected the mentioned errors and revised the references.
References 27-33 do not refer to gastric cancer. Please provide appropriate references on gastric cancer
Answer: we have revised and replaced the mentioned references.
Lines 194-197 Esophagus carcinoma is out of context here.
Answer: we have deleted the example of esophagus carcinoma for clarity.
Reference numbers in table 1 are not correct.
Answer: we have corrected relevant references.
Lines 275-277. This sentence is not clear.
Answer: we have reworded the sentence.
Lines 290-291 “FDG PET/CT, as a newly developed imaging tool, is well known for HNCs staging and treatment”. Please reword this sentence.
Answer: we have reworded the sentence.
Line 295 “HNCs under the CAFs category overexpress FAPI” What does under the CAF category mean?
Answer: we have reworded the sentence for clarity.
Line 298-300 “HNCs has been extensively studied by several research groups” Please provide references.
Answer: we have included relevant references.
Line 301-303 The study by Syed was incorrectly reported since this author studied 14 patients with head and neck cancer.
Answer: we have corrected the description.
Lines 306-312. It is not clear how FAPI can sensitize whole tumor to RT. Also the sentence “FAPI PET/CT also improves surveillance of therapeutic management via dose painting, which is achieved based on the SUVs” is not clear.
Answer: we have included an explanation for how FAPI could sensitize the tumor to RT, and we have also deleted the mentioned sentence for clarity.
Lines 321-324 Please reword this sentence.
Answer: we have reworded the sentence.
Line 333-335. “68Ga-FAPI PET/CT surpasses 18F-FDG PET/CT in the diagnosis and staging of lung cancer and is applied in the detection of early-stage lung cancer”. Please specify that 68Ga-FAPI PET/CT is not currently included in the work-up of lung cancer patients.
Answer: we have specified that FAPI imaging is currently not a part of the standard examination for patients with lung cancer.
Line 340-341 “SUVmax was similar but TBG was higher” Where?
Answer: we have specified the location of higher SUVmax and TBR by FAPI.
In addition to the points raised, paragraphs 2.3, 2.4, 2.6 and 2.7 should be carefully revised to improve the content and the English. Also the concept of using PET images for delineation of gross tumor volume in radiotherapy planning should be better explained.
Answer: we have revised the mentioned paragraphs, and included an explanation for how FAPI helps the delineation of GTVs for RT planning.

Reviewer 2 Report
There are still quite a lot of linguistic, grammatical and typo errors in the work, which sometimes makes it difficult to understand the author's thoughts. There are so many minor errors that it is impossible to list them all review, for example: in title Malignacies, should be: malignacies; line 29 ac-id-based (acid-based) and others.
The summary of the work is very general and does not contain any new conclusions or suggestions.
In a previous review, in advance, I gave a slightly higher score due to the interesting subject matter and potential improvement in the quality of the manuscript. Unfortunately, at the moment I think that despite the corrections made, the overall quality of the work has not improved.
The work requires, first of all, a thorough linguistic correction.
Author Response
There are still quite a lot of linguistic, grammatical and typo errors in the work, which sometimes makes it difficult to understand the author's thoughts. There are so many minor errors that it is impossible to list them all review, for example: in title Malignacies, should be: malignacies; line 29 ac-id-based (acid-based) and others.
The summary of the work is very general and does not contain any new conclusions or suggestions.
In a previous review, in advance, I gave a slightly higher score due to the interesting subject matter and potential improvement in the quality of the manuscript. Unfortunately, at the moment I think that despite the corrections made, the overall quality of the work has not improved.
The work requires, first of all, a thorough linguistic correction.
Answer: Thank you for your time and effort in reviewing this manuscript. To emphasize our points and improve lingual quality, we have carefully revised the manuscript and obtained English revision for native speakers. We are willing to submit proof along with the revised manuscript for your reference.

Round 3
Reviewer 1 Report
The authors revised the manuscript taking into account each point raised.
Reviewer 2 Report
After the corrections made by the authors, the work is more coherent and more readable.
Visible minor inconsistencies do not diminish the substantive value of the work.